# Topology-induced symmetry breaking demonstrated in antiferromagnetic magnons on a Möbius strip

**Kuangyin Deng[1][*] and Ran Cheng[1,2,3][†]**

**1** Department of Electrical and Computer Engineering, University of California, Riverside, California 92521, USA
**2** Department of Physics and Astronomy, University of California, Riverside, California 92521, USA
**3** Department of Materials Science and Engineering, University of California, Riverside, California 92521, USA

[*] kuangyid@ucr.edu ,   [†] rancheng@ucr.edu

## Abstract

We propose a mechanism of topology-induced symmetry breaking, where certain local symmetry preserved by the Hamiltonian is explicitly broken in the eigenmodes of excitation due to nontrivial real-space topology. We demonstrate this phenomenon by studying magnonic excitations on a Möbius strip comprising two antiferromagnetically coupled spin chains. Even with a simple Hamiltonian respecting local rotational symmetry, with all local curvature effects ignored, magnons exhibit linear polarization of the Néel vector devoid of chirality, forming two non-degenerate branches that cannot be smoothly connected to nor globally decomposed into, the circularly-polarized magnons. Correspondingly, one branch undergoes a spectral shift and only admits standing waves of half-integer wavelength, whereas the other only affords standing waves of integer wavelength. Under the Möbius boundary condition, we further identify an exotic phase hosting spontaneous antiferromagnetic order whilst all exchange couplings are ferromagnetic. The suppression of chirality in the order parameter dynamics, hence the pattern of standing waves, can be generalized to other elementary excitations on non-orientable surfaces. Our findings showcase the profound influence of real-space topology on the physical nature of not just the ground state but also the quasiparticles.

## 1  Introduction

### 1.1  Background

In solid-state systems, symmetry and interactions can directly manifest in the physical properties of quasiparticles (*i.e.*, quanta of elementary excitations) by affecting the momentum-space topology, whereas the subtle impact of real-space topology remains elusive. Prevailing studies customarily adopt periodic boundary conditions (PBCs) in real space when dealing with quasiparticles [1], for which the system becomes topologically equivalent to a circle, a torus, or a 3D-torus depending on the dimensionality. However, there exist exotic structures such as Möbius strips that inherently do not conform to the PBCs and cannot be smoothly deformed into tori. Figure 1 illustrates that a Möbius strip is non-orientable by nature as it consists of a single surface and a single edge, which leads to an ambiguous *global* normal vector, precluding the validity of ordinary PBCs.

Concerning the physical behavior of quasiparticles on such a non-orientable object and their subtle relations with the ground state, it is tempting to ask: what are the implications of the topologically non-trivial boundary conditions? Recently, quasiparticles residing on Möbius strips aroused increasing theoretical attentions [2–9]. On the experimental side, Möbius strips have been realized in a wide variety of systems such as molecules [10,11], single crystals [12], resonators [13–15], and optical cavities [16,17], fertilizing a vibrant arena for exploring new physics emerging from the Möbius topology. Nevertheless, most of these studies focused on the local curvature effects under continuous geometry. It is far from clear if there exist any residual consequences arising *only* from the Möbius boundary condition when spatial curvature is discarded, which can even survive in the limit of very large systems.

### 1.2  General Considerations

A direct consequence following the ambiguity of global normal vector is the absence of a globally defined chirality. As illustrated in Fig. 1, the right-handed direction *locally* associated with the normal vector inevitably conflicts with itself as we loop around the Möbius strip for a full cycle, fundamentally disrupting the eigenmodes of excitations of all kinds accompanied by chirality. In particular, it is impossible to distinguish between the right-handed and left-handed polarizations.

To be more quantitative, let us construct a local orthogonal basis shown in Fig. 1, wherein $z$ coincides with the local normal vector while $x$ ($y$) is along the longitudinal (transverse) direction of the strip. We represent the linearly polarized modes along $x$ and $y$ by $|\phi_x(\boldsymbol{r})\rangle$ and $|\phi_y(\boldsymbol{r})\rangle$, respectively, with $\boldsymbol{r} = \{x, y\}$ specifying the location on the strip. Through gauge fixing, $|\phi_x(\boldsymbol{r})\rangle$ and $|\phi_y(\boldsymbol{r})\rangle$ can be locally in phase. Generally, a mode with chirality is expressed as a coherent superposition:

$$|\psi\rangle = \sum_{\boldsymbol{r}} \mathcal{N}(\boldsymbol{r}) \big[ |\phi_x(\boldsymbol{r})\rangle + f(\boldsymbol{r}) e^{i\theta(\boldsymbol{r})} |\phi_y(\boldsymbol{r})\rangle \big], \tag{1}$$

where $\mathcal{N}(\boldsymbol{r})$ is a normalization factor; $f(\boldsymbol{r})$ and $\theta(\boldsymbol{r})$ are independent real continuous functions. At $\boldsymbol{r}$, $|\psi\rangle$ exhibits the right-handed (left-handed) chirality for $\theta(\boldsymbol{r}) < 0$ ($> 0$). The corresponding classical trajectory is elliptical for $\theta(\boldsymbol{r}) \neq \pm\pi/2$, with the principal axes along

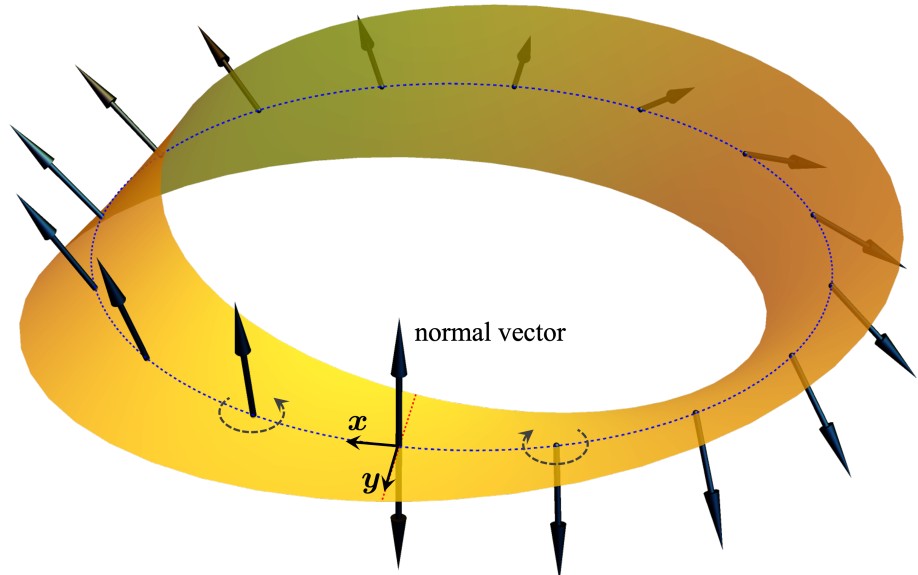

Figure 1: A Möbius strip is non-orientable because the normal vector cannot be globally defined and the PBCs are inapplicable. As a result, chirality associated with the normal vector (illustrated in dashed arrows) becomes ambiguous, rendering the chiral modes of elementary excitations (such as circularly polarized magnons) at odds with the boundary condition.

$\tan^{-1} f(\boldsymbol{r})$ and $\tan^{-1} f(\boldsymbol{r}) + \pi/2$. When $\theta(\boldsymbol{r}) = \pm\pi/2$, the trajectory is circular and all directions can be viewed as principal axes. When $\theta(\boldsymbol{r}) \to 0$, the trajectory shrinks to a line so that the polarization becomes linear. In the summation over $\boldsymbol{r}$, $x$ runs from 0 to $L$ with $L$ labeling the length of the Möbius strip, while $y$ runs over the width from $-w/2$ to $+w/2$. Should $\theta(\boldsymbol{r})$ be allowed to keep the same sign everywhere on the strip, the mode represented by $|\psi\rangle$ will acquire a globally well-defined chirality.

To impose the boundary conditions, we define $\hat{T}_L$ as the translation operator (along $x$) that moves functions of $\boldsymbol{r}$ from $(x, y)$ to $(x + L, y)$ by continuously sliding the local coordinate frame along the strip. Since the linearly polarized modes are not entitled with chirality *in priori* and are not disrupted by the ambiguity of the normal vector, they must be single valued, thus

$$\hat{T}_L |\phi_x(x,y)\rangle = |\phi_x(x+L,y)\rangle = |\phi_x(x,-y)\rangle, \tag{2a}$$

$$\hat{T}_L |\phi_y(x,y)\rangle = |\phi_y(x+L,y)\rangle = |\phi_y(x,-y)\rangle. \tag{2b}$$

On the other hand, the superposition coefficients satisfy

$$\hat{T}_L \mathcal{N}(x,y) = \mathcal{N}(x+L,y) = \mathcal{N}(x,-y), \tag{3a}$$

$$\hat{T}_L f(x,y) = f(x+L,y) = f(x,-y), \tag{3b}$$

$$\hat{T}_L \theta(x,y) = \theta(x+L,y) = -\theta(x,-y), \tag{3c}$$

where $-1$ in the last equation is interpreted as $e^{\pm i\pi}$ with $\pm$ referring to the two distinct ways of wrapping up the Möbius strip [see the schematics in Fig. 2(b) and (c)]. In general, if a strip is multiply twisted when attaching its two ends, $\hat{T}_L \theta[x, y] = \exp(2ip\pi)\theta[x, (-1)^{2|p|} y]$ where $p$ registers the number of turns on the strip. The Möbius strip is half-twisted so $p = \pm 1/2$.

If $|\psi\rangle$ serves as an eigenmode of excitation, it must be single valued, *i.e.*, invariant after traversing the Möbius strip by $L$, or simply $\hat{T}_L |\psi\rangle = |\psi\rangle$. Applying the above boundary

conditions, we obtain

$$\hat{T}_L |\psi\rangle = \sum_r \mathcal{N}(x,-y)\Big[|\phi_x(x,-y)\rangle + f(x,-y)e^{-i\theta(x,-y)}|\phi_y(x,-y)\rangle\Big]$$

$$= \sum_{r'} \mathcal{N}(r')\Big[|\phi_x(r')\rangle + f(r')e^{-i\theta(r')}|\phi_y(r')\rangle\Big], \tag{4}$$

where the symmetric range of summation over $y$ has been exercised when $r'$ replaces $r$ in the last step. Comparing Eq. (4) with Eq. (1), we have $\hat{T}_L |\psi\rangle = |\psi\rangle$ if and only if $\theta(r)$ vanishes identically, indicating that $|\psi\rangle$ is strictly linearly-polarized. That is to say, a single-valued mode on the Möbius strip cannot possess a definite chirality, whereas a mode with prescribed chirality cannot be single valued, let alone being an eigenmode.

The Möbius topology does not place a strong restriction on the Hamiltonian $H$, so long as $H$ is single valued and respects $\hat{T}_L H(x,y) = H(x+L,y) = H(x,-y)$. For instance, the Heisenberg Hamiltonian acting only in the spin space preserves the local spin-rotational symmetry, not contradicting with the boundary conditions. In the topological trivial scenarios, this symmetry ensures the existence of circularly-polarized eigenmodes in the spin excitations, which are furnished by well-defined chirality. On the other hand, the Möbius topology can drastically modify these eigenmodes and suppress their chirality, as analyzed above. At this point, it is inspiring to define a hitherto overlooked mechanism, which we name as topology-induced symmetry breaking (TISB):

*Certain local symmetry preserved by the Hamiltonian is explicitly broken in the eigenmodes of excitation (or quasiparticles) owing to the real-space topology.*

This mechanism unravels the extraordinary behavior of quasiparticles exclusively enabled by the topological boundary conditions (TBCs) but has nothing to do with the local curvature effect. One should not confuse TISB with the well-known mechanism of *spontaneous symmetry breaking*, in which a symmetric Hamiltonian selects an asymmetric *ground state*—thereby concealing the broken symmetry in its Goldstone modes. While TISB could entail profound consequences in various physical systems with non-trivial real-space topology, in this paper we demonstrate its manifestation in the magnonic excitations on a Möbius strip.

## 1.3   System Description

To avoid confusion with spontaneous symmetry breaking associated with the ground state, here we consider a special case where the ground state remains intact and trivial, whereas the eigenmodes of excitation are fundamentally disrupted by the Möbius topology. As illustrated in Fig. 2(a), we consider a nano-ribbon composed of two ferromagnetic spin chains that are oppositely aligned, forming an effective antiferromagnetic (AFM) system. The nano-ribbon can form a Möbius strip in two distinct ways according to Fig. 2(b) and (c). To motivate our study, we first make a critical observation: although the AFM configuration in the ground state is compatible with the Möbius TBC, the magnon excitations with circular polarizations are inherently irreconcilable with the system topology. To be specific, when only the exchange interactions and the easy-axis anisotropy are considered, the spin Hamiltonian respects the local $O(2)$ rotational symmetry in the spin space. Consequently, if the TBC is disregarded, the magnon eigenmodes would become circularly polarized, exhibiting either left-handed or right-handed chirality for both spin species [18–22]. For example, in the left-handed mode depicted in Fig. 2(a), $S_A$ and $S_B$ both rotate clockwisely but $S_A$ has a larger oscillation amplitude. However, as illustrated in Fig. 2(b) and (c), imposing the Möbius TBC by connecting the 1 and $N$ sites (with A and B swapped) will inevitably disrupt both the chirality and the amplitude of the spin precessions. Therefore, the magnon solutions that are commensurate with the Möbius topology must be fundamentally different from the circularly-polarized modes widely known for collinear AFM materials.

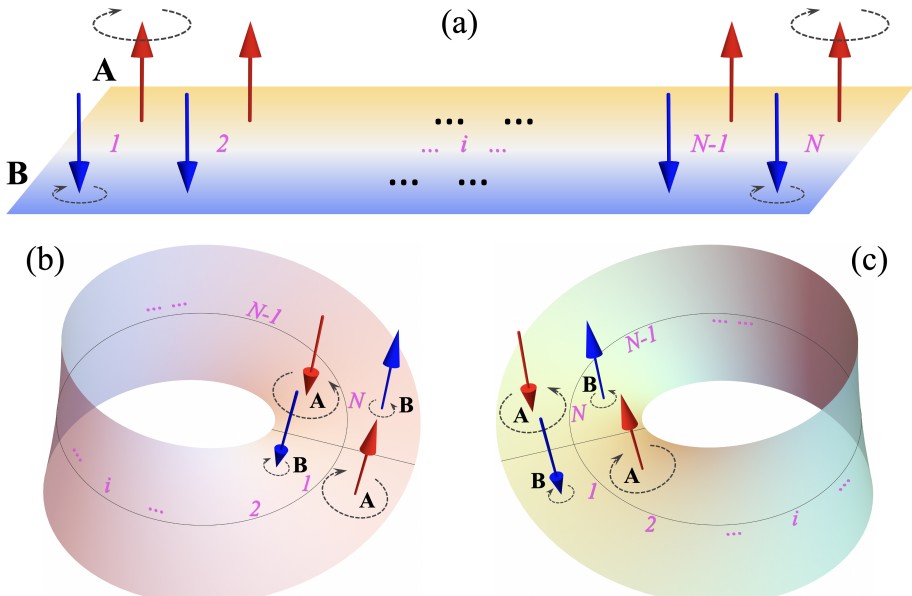

Figure 2: Schematic illustration of the system. (a) An AFM nano-ribbon consists of two ferromagnetic spin chains, where the red (blue) arrows signify the equilibrium spin orientation of the A (B) sub-lattice. The black dashed arrows indicate the manner of spin precessions associated with the left-handed magnon mode, where $S_A$ has a larger amplitude than $S_B$. On the contrary, the right-handed mode is characterized by a larger precession of $S_B$ over $S_A$ (not shown). (b) and (c) depict the two distinct ways of connecting the ribbon into a Möbius strip. While the AFM ground state is compatible with the TBC, the excited states in the form of circularly-polarized magnons are radically disrupted.

## 2   Model and Solutions

Consider a nano-ribbon with the AFM configuration shown in Fig. 2(a), where the red and blue arrows indicate the spins on the A and B sub-lattices in the ground state. A Möbius strip can then be constructed based on either Fig. 2(b) or Fig. 2(c), which are topologically distinct and satisfy different TBCs to be specified later. To unambiguously separate the influence of real-space topology from local geometric effects, we intentionally exclude the curvature of the strip [23–29]. In this regard, we adopt a minimal Hamiltonian that preserves the local rotational symmetry about the local $z$ axis:

$$H_0 = -J_F \sum_{\langle i,j \rangle} (S_{Ai} \cdot S_{Aj} + S_{Bi} \cdot S_{Bj}) + J_{AF} \sum_i S_{Ai} \cdot S_{Bi} - K \sum_i (S_{Ai}^{z\,2} + S_{Bi}^{z\,2}), \qquad (5)$$

where $i$ labels the lattice on the strip, $J_F$ ($J_{AF}$) is the nearest-neighbor exchange interaction for (between) the same (different) spin species, and $K$ is the perpendicular easy-axis anisotropy. In our convention, all parameters are positive. The collinear AFM ordering depicted in Fig. 2 is well protected by the magnetic anisotropy $K$, which suppresses quantum fluctuations in the reduced dimension. The summation $\langle i, j \rangle$ is equivalent on all links, including that between 1 and $N$, such that every link in the longitudinal direction looks the same; we are free to renumber the whole strip by moving 1 to anywhere. Contrary to a single ferromagnetic spin chain arranged on a Möbius strip [23, 24], our system is free from geometrical frustration, thus no domain wall is present.

    To derive the quantum magnon excitations, we apply the linearized Holstein–Primakoff transformations on the spin raising and lowering operators (assuming that $S$ is sufficiently

large), $S^{\pm} = S^x \pm iS^y$, for each sublattice

$$S_{Ai}^{+} \approx \sqrt{2S}a_i, \quad S_{Ai}^{-} \approx \sqrt{2S}a_i^{\dagger}, \quad S_{Ai}^{z} = S - a_i^{\dagger}a_i, \tag{6a}$$

$$S_{Bi}^{+} \approx \sqrt{2S}b_i^{\dagger}, \quad S_{Bi}^{-} \approx \sqrt{2S}b_i, \quad S_{Bi}^{z} = b_i^{\dagger}b_i - S, \tag{6b}$$

where $a_i$ ($b_i$) represents the annihilation of a magnon on site $i$ and sublattice A (B), and $S$ is the spin magnitude on each site. By neglecting the constant terms, we obtain the magnon Hamiltonian as

$$H = (2K + 2J_F + J_{AF})S \sum_i (a_i^{\dagger}a_i + b_i^{\dagger}b_i) - J_F S \sum_{\langle i,j \rangle} (a_i^{\dagger}a_j + a_j^{\dagger}a_i + b_i^{\dagger}b_j + b_j^{\dagger}b_i)$$
$$+ J_{AF}S \sum_i (a_i^{\dagger}b_i^{\dagger} + a_i b_i). \tag{7}$$

We cannot directly apply Fourier transformations to Eq. (7) because the PBCs, $a_{i+N} = a_i$ and $b_{i+N} = b_i$, are explicitly broken. Instead, we have the TBCs

$$a_{i+N} = b_i, \qquad b_{i+N} = a_i, \tag{8}$$

which means the definitions of A and B chains are interchanged after winding around the Möbius strip. Basing on Eq. (8), we can recombine $a_i$ and $b_i$ as

$$\alpha_i = \frac{1}{\sqrt{2}}(a_i - b_i)e^{i\xi\frac{\pi x_i}{L}}, \qquad \beta_i = \frac{1}{\sqrt{2}}(a_i + b_i), \tag{9}$$

where $x_i$ specifies the position of site $i$ along the strip [see Fig. 2], $L$ is the total length of the nano-ribbon, and $\xi = \pm 1$ corresponds to the two distinct ways of connection illustrated in Fig. 2(b) and 2(c). Now, the new operators $\alpha_i$ and $\beta_i$ satisfy not only the bosonic commutation relations but also the PBCs

$$\alpha_{i+N} = \alpha_i, \qquad \beta_{i+N} = \beta_i. \tag{10}$$

Under this new set of basis, Equation (7) becomes

$$H = (2K + 2J_F + J_{AF})S \sum_i \left[ \alpha_i^{\dagger}\alpha_i + \beta_i^{\dagger}\beta_i \right] - J_F S \sum_{\langle i,j \rangle} \left[ e^{i\pi\xi(x_i - x_j)/L}\alpha_i^{\dagger}\alpha_j + \beta_i^{\dagger}\beta_j + h.c. \right]$$
$$- \frac{J_{AF}S}{2} \sum_i \left[ e^{i2\pi\xi x_i/L}\alpha_i^{\dagger}\alpha_i^{\dagger} - \beta_i^{\dagger}\beta_i^{\dagger} + h.c. \right], \tag{11}$$

where $h.c.$ denotes hermitian conjugate. Equation (11) is naturally decomposed into $H = H_\alpha + H_\beta$ for the $\alpha_i$ and $\beta_i$ sectors. Applying the Fourier transformations

$$\alpha_k = \frac{1}{\sqrt{2N}} \sum_i e^{-i(k-\xi\pi/L)x_i}(a_i - b_i), \tag{12a}$$

$$\beta_k = \frac{1}{\sqrt{2N}} \sum_i e^{-ikx_i}(a_i + b_i), \tag{12b}$$

we can derive the momentum-space Hamiltonian. To this end, we adopt the Bogoliubov-de-Gennes (BdG) basis

$$\Psi_\alpha = (\alpha_k, \alpha_{-k+2\pi\xi/L}^{\dagger})^{\mathrm{T}}, \qquad \Psi_\beta = (\beta_k, \beta_{-k}^{\dagger})^{\mathrm{T}} \tag{13}$$

with $l = L/N$ being the lattice constant, $H_\alpha = \Psi_\alpha^{\dagger}\mathcal{H}_\alpha\Psi_\alpha$ and $H_\beta = \Psi_\beta^{\dagger}\mathcal{H}_\beta\Psi_\beta$. Here, the BdG Hamiltonian reads

$$\mathcal{H}_\alpha = S \begin{pmatrix} Q_\alpha & -J_{AF}/2 \\ -J_{AF}/2 & Q_\alpha \end{pmatrix}, \qquad \mathcal{H}_\beta = S \begin{pmatrix} Q_\beta & J_{AF}/2 \\ J_{AF}/2 & Q_\beta \end{pmatrix}, \tag{14}$$

where $Q_\alpha = K + J_{AF}/2 + J_F[1 - \cos(k - \xi\pi/L)l]$ and $Q_\beta = K + J_{AF}/2 + J_F[1 - \cos(kl)]$. It should be noted that in the $\alpha$ sector, magnons of momentum $k$ couple those of momentum $-k + 2\pi\xi/L$; whereas in the $\beta$ sector, $k$ pairs with $-k$ without a shift.

Owing to the bosonic commutation relations of the BdG basis, we need to diagonalize $\sigma_z \mathcal{H}_{\alpha(\beta)}$ rather than $\mathcal{H}_{\alpha(\beta)}$ for the magnon solutions [30]. This can properly take care of the "minus sign" associated with the Bogoliubov normalization for bosons [31]. By doing so, we obtain the eigen-frequencies (we set $\hbar = 1$)

$$\omega_{\alpha(\beta)}^\pm = \pm S \sqrt{q_{\alpha(\beta)}^1 q_{\alpha(\beta)}^2} \tag{15}$$

with the corresponding eigenvectors (unnormalized)

$$v_\alpha^\pm = \left( \sqrt{q_\alpha^1} \pm \sqrt{q_\alpha^2}, \sqrt{q_\alpha^1} \mp \sqrt{q_\alpha^2} \right)^{\mathrm{T}}, \tag{16a}$$

$$v_\beta^\pm = \left( \sqrt{q_\beta^1} \pm \sqrt{q_\beta^2}, -\sqrt{q_\beta^1} \pm \sqrt{q_\beta^2} \right)^{\mathrm{T}}, \tag{16b}$$

where $q_{\alpha(\beta)}^1 = Q_{\alpha(\beta)} + J_{AF}/2$ and $q_{\alpha(\beta)}^2 = Q_{\alpha(\beta)} - J_{AF}/2$ are both positive. The negative frequency branches and their associated eigenvectors are redundant solutions, which can be interpreted as a *hole* representation. For instance, $v_\beta^-(k)$ describes a hole at $k$ that corresponds to a real $\beta$-magnon at $-k$. A similar picture is applicable to the $\alpha$ branch so long as the $2\pi\xi/L$ momentum shift appearing in Eq. (13) is taken into account. Consequently, $v_{\alpha(\beta)}^+$ and $v_{\alpha(\beta)}^-$ are linearly dependent, representing one unique physical solution.

Let us concentrate on the positive frequency branches and consider the $\xi = 1$ connection [*i.e.*, Fig. 2(b)]. For simplicity, we also omit the super-index $+$. With a proper normalization of $v_\alpha^+$ and $v_\beta^+$, the magnon eigenmodes associated with $\omega_\alpha$ and $\omega_\beta$ are described respectively by

$$\tilde{\alpha}_k = \frac{\sqrt{q_\alpha^1} + \sqrt{q_\alpha^2}}{2(q_\alpha^1 q_\alpha^2)^{\frac{1}{4}}} \alpha_k + \frac{\sqrt{q_\alpha^1} - \sqrt{q_\alpha^2}}{2(q_\alpha^1 q_\alpha^2)^{\frac{1}{4}}} \alpha_{-k+2\pi/L}^\dagger \tag{17a}$$

$$\tilde{\beta}_k = \frac{\sqrt{q_\beta^1} + \sqrt{q_\beta^2}}{2(q_\beta^1 q_\beta^2)^{\frac{1}{4}}} \beta_k - \frac{\sqrt{q_\beta^1} - \sqrt{q_\beta^2}}{2(q_\beta^1 q_\beta^2)^{\frac{1}{4}}} \beta_{-k}^\dagger \tag{17b}$$

and their counterparts $\tilde{\alpha}_k^\dagger$, $\tilde{\beta}_k^\dagger$. Figure 3(a) and (b) plot the discretized dispersion relations for $N = 10$ (only the lowest few states on each branch are shown), along with illustrations of the magnon eigenmodes at $k = 0$. While the $\beta$ modes distribute symmetrically $\omega_\beta(-k) = \omega_\beta(k)$, the $\alpha$ branch shifts rightward by $\delta k = \pi/L$ such that $\omega_\alpha(-k) = \omega_\alpha(k + 2\delta k)$. The skewed $\omega_\alpha(k)$ is intimately related to the asymmetric paring of $\Psi_\alpha$ in Eq. (13), which originates from the non-trivial topology of the Möbius strip. It is easy to verify that setting $\xi = -1$ [*i.e.*, using the connection of Fig. 2(c)] leads to a leftward shift of $\omega_\alpha(k)$, or $\delta k = -\pi/L$. Interestingly, if we reversely count the sites on the strip, the spectral shift $\delta k$ also flips sign, but in this case the eigenvectors are different from what one would obtain for $\xi = -1$.

To intuitively understand the magnon eigenmodes, we now express Eq. (17) in terms of the original spin variables. Using Eqs. (6) and (12), we obtain

$$\tilde{\alpha}_k^\dagger = \sum_i \frac{e^{i(k-\delta k)x_i}}{2(q_\alpha^1 q_\alpha^2)^{\frac{1}{4}} \sqrt{NS}} \left[ \sqrt{q_\alpha^1} S_{Ai}^x - i\sqrt{q_\alpha^2} S_{Ai}^y - \sqrt{q_\alpha^1} S_{Bi}^x - i\sqrt{q_\alpha^2} S_{Bi}^y \right], \tag{18a}$$

$$\tilde{\beta}_k^\dagger = \sum_i \frac{e^{ikx_i}}{2(q_\beta^1 q_\beta^2)^{\frac{1}{4}} \sqrt{NS}} \left[ \sqrt{q_\beta^2} S_{Ai}^x - i\sqrt{q_\beta^1} S_{Ai}^y + \sqrt{q_\beta^2} S_{Bi}^x + i\sqrt{q_\beta^1} S_{Bi}^y \right]. \tag{18b}$$

The classical limit of Eqs. (6) can be established by considering the quantum averages: $\langle S_A^- \rangle = \sqrt{2S}\langle a^\dagger \rangle$ and $\langle S_B^+ \rangle = \sqrt{2S}\langle b^\dagger \rangle$ correspond to the left-handed precessions of $S_A$ and $S_B$ with respect to the equilibrium direction of $S_A$ (the reference direction). By a straightforward algebra, we obtain the real-time evolution of the classical spin vectors as

$$S_{A/B}^\alpha \sim \mathrm{Re}\left[\left(\pm\sqrt{q_\alpha^1}\hat{x} + \mathrm{i}\sqrt{q_\alpha^2}\hat{y}\right)e^{\mathrm{i}\omega_\alpha t - \mathrm{i}(k-\delta k)x}\right], \tag{19a}$$

$$S_{A/B}^\beta \sim \mathrm{Re}\left[\left(\sqrt{q_\beta^2}\hat{x} \pm \mathrm{i}\sqrt{q_\beta^1}\hat{y}\right)e^{\mathrm{i}(\omega_\beta t - kx)}\right], \tag{19b}$$

for the $\alpha$- and $\beta$-branch, respectively, where the $+$ $(-)$ sign corresponds to the A (B) sublattice. Since $q_\alpha^1 > q_\alpha^2 > 0$ and $q_\beta^1 > q_\beta^2 > 0$, both branches exhibit elliptical precession: $S_A$ rotates left-handedly while $S_B$ rotates right-handedly, with the $\alpha$ mode's major axis along $x$ and the $\beta$ mode's along $y$; $S_A$ and $S_B$ always precess about the easy-axis with the same amplitude but opposite chirality. As a result, the Néel vector $n = (S_A - S_B)/2S$ exhibits linear oscillation devoid of chirality, on which we will further elaborate in the next section.

As demonstrated in Fig. 3(a)–(b), the two branches are quite different in character even though they share one thing in common: from a local bird-eye view, $S_A$ and $S_B$ in both $\alpha$ and $\beta$ mode at $k = 0$ overlap with each other when passing the minor axes of their elliptical trajectories while becoming back-to-back when passing the major axes. Due to the momentum shift $\delta k = \pi/L$ in the $\alpha$ branch, the spin precessions on site $i = N+1$ (same as $i = 1$) acquires a $\pi$ phase (accumulated through every site) even for $k = 0$, which exactly compensates the impact of the Möbius TBC that connects sites 1 and $N$ with a half twist. In contrast, the $\beta$ mode at $k = 0$ does not exhibit any phase difference between 1 and $N$, which is just commensurate with the TBC. From a local perspective (disregarding the phase difference among different sites), the $\alpha$ and $\beta$ modes have their major axes of spin precessions perpendicular to each other, rendering the planes of Néel vector oscillation orthogonal.

## 3 Analysis and Discussion

To further understand the unique characteristics of the magnon eigenmodes obtained above, we draw a 3D perspective in Fig. 3(c) where $S_A$ and $S_B$ share the same origin such that their precessional trajectories are concentric about the local $z$ axis. It is easy to deduce that the Néel vector $n = (S_A - S_B)/2S$ undergoes a pendulum-like oscillation confined on the plane containing the major axes of the two elliptical trajectories. Comparatively, the total spin vector $S = (S_A + S_B)/2S$ oscillates linearly on a plane orthogonal to that of $n$. Per the definition of spin wave polarization (in terms of the order parameter dynamics) [19–21], both the $\alpha$ and $\beta$ modes are considered linearly polarized, thus being devoid of chirality.

The above intuitive picture can be corroborated by Eq. (19), from which we can further read off the oscillating components of the Néel vector and the total spin vector as

$$\delta n_\alpha(k) \sim \hat{x}\sqrt{q_\alpha^1}\cos\left[\omega_\alpha(k)t - (k-\delta k)x\right], \tag{20a}$$

$$S_\alpha(k) \sim \hat{y}\sqrt{q_\alpha^2}\sin\left[\omega_\alpha(k)t - (k-\delta k)x\right], \tag{20b}$$

$$\delta n_\beta(k) \sim \hat{y}\sqrt{q_\beta^1}\sin\left[\omega_\beta(k)t - kx\right], \tag{20c}$$

$$S_\beta(k) \sim \hat{x}\sqrt{q_\beta^2}\cos\left[\omega_\beta(k)t - kx\right], \tag{20d}$$

all of which are indeed linearly polarized bearing null chirality, confirming the picture inferred in Fig. 3(c). As $q_{\alpha(\beta)}^1 > q_{\alpha(\beta)}^2 > 0$, the oscillation amplitude of the Néel vectors are larger than that of the total spin in both modes, namely, $|\delta n_{\alpha(\beta)}| > |S_{\alpha(\beta)}|$. The geometrical relations embedded in Eq. (20) are illustrated by the right panel of Fig. 3(c).

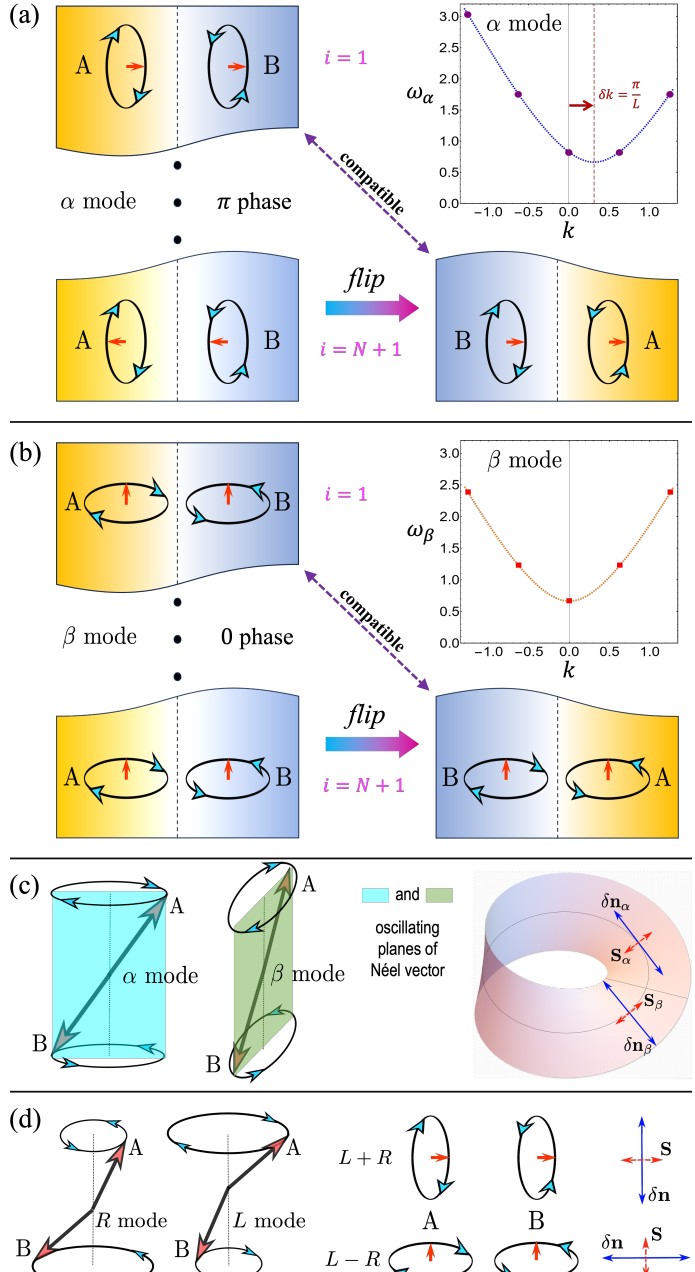

Figure 3: (a, b) Illustrations of the $\alpha$ and $\beta$ modes at $k = 0$, and the plots of the lowest few modes on each branch for $\xi = 1$, $N = 10$, $l = 1$, $S = 2$, $J_F = J_{AF} = 1$, and $K = 0.1$. While $\omega_\beta(-k) = \omega_\beta(k)$ is symmetric, $\omega_\alpha(-k) = \omega_\alpha(k + 2\delta k)$ is skewed by $\delta k = \pi/L$. The spectral shift of the $\alpha$ branch is accompanied by a cumulative $\pi$ phase in the spin precessions as we move from site $i$ to $i + N$ (*i.e.*, traveling by a full loop), reflecting the impact of the Möbius TBC. (c) Left: a 3D illustration of the spin precessions in the $\alpha$ and $\beta$ modes, where $S_A$ and $S_B$ rotate elliptically with an equal amplitude and opposite chirality, rendering the Néel vector linearly-polarized. Right: illustration of the oscillating Néel vector (solid blue) and the oscillating spin vector (dashed red) for the $\alpha$ and $\beta$ modes based on Eq. (20). That $\delta n_\alpha \perp \delta n_\beta$ should not be confused by their being shown in different locations. (d) Ordinary AFM right-circular ($R$) and left-circular ($L$) magnon modes, and their superposition $L + R$ and $L - R$ characterized by a linearly-polarized Néel vector (solid blue) orthogonal to the oscillating spin vector (dashed red).

The total number of sites $N = L/l$ is finite, so the magnon eigenmodes must be discrete, taking place only at $k = 0, \pm 2\pi/L, \pm 4\pi/L \cdots$ as plotted in Fig. 3. Because the spectrum $\omega_\alpha(k) \neq \omega_\alpha(-k)$ is skewed by $\delta k = \pi/L$ while $\omega_\beta(k) = \omega_\beta(-k)$ is symmetric, the formation of standing waves by these two branches is quite different. A standing wave features separation of $t$ and $x$ in the wavefunction, which, under the boundary conditions, restricts the way of mode pairing. For the $\alpha$ branch, a pair of magnons of momenta $k_{n\pm}^\alpha = \delta k(1 \pm n)$ can superimpose and form a standing wave

$$\delta \boldsymbol{n}_\alpha(k_{n+}^\alpha) + \delta \boldsymbol{n}_\alpha(k_{n-}^\alpha) \sim \hat{x} \cos(\omega_\alpha t) \cos(n\delta kx), \tag{21}$$

where $n = 1, 3, 5...$ is odd integer representing the number of nodes as shown in Fig. 4(a). For an even integer, it is impossible to separate $t$ and $x$ because of the spectral shift and its ensuing $\pi$-phase shift in the spin precessions. On the other hand, mode pairing within the $\beta$ branch can only happen between $k_{n\pm}^\beta = \pm n\delta k$ with an even integer $n = 0, 2, 4...$ such that

$$\delta \boldsymbol{n}_\beta(k_{n+}^\beta) + \delta \boldsymbol{n}_\beta(k_{n-}^\beta) \sim \hat{y} \sin(\omega_\beta t) \cos(n\delta kx), \tag{22}$$

which is shown in Fig. 4(a). Comparatively, the $\beta$ branch bears a similar standing-wave formation as the case of trivial PBC, whereas the $\alpha$ branch is unconventional and unique to the TBC. While the number of nodes in each standing-wave mode is well-defined, their locations indicated by the $\cos(n\delta kx)$ factor in Eqs. (21) and (22) are determined up to a global shift. This is because the Hamiltonian Eq. (5) is translationally invariant (with respect to a relocation of $i = 1$).

At this point, it is instructive to compare the unique magnon eigenmodes on a Möbius strip with what would become the eigenmodes if the topologically trivial PBC is imposed on the nano-ribbon in Fig. 2(a). Under the PBC, the nano-ribbon only admits the well-known eigenmodes in collinear AFM materials [19–21] since we have excluded the local curvature effect from the Hamiltonian. As illustrated in Fig. 3(d), these modes are the right-circular ($R$) and left-circular ($L$) eigenmodes, which, by a coherent superposition, can form $L + R$ and $L - R$ featuring elliptical precessions of $\boldsymbol{S}_A$ and $\boldsymbol{S}_B$ with opposite chirality, leading to a linearly-polarized oscillation of the Néel vector $\boldsymbol{n}$ with null chirality (so is $\boldsymbol{S}$). While $L + R$ and $L - R$ are linearly polarized, the eigenspace they span is topologically distinct from that spanned by the $\alpha$ and $\beta$ modes. To be specific, while $L + R$ is locally identical to the $\alpha$ mode, it is not accompanied by a built-in $\pi/L$ phase shift at $k = 0$ for every location, let alone a spectral shift. Despite their superficial similarity to $(L - R)$ and $(L + R)$, the $\alpha$ and $\beta$ modes cannot be exactly expressed as a linear superposition of $R$ and $L$. This is because the Möbius strip is a non-orientable manifold on which the chirality of spin precession becomes ambiguous globally, given that $+z$ and $-z$ are indistinguishable. In other words, the eigenspace spanned by $\alpha$ and $\beta$ is not smoothly connected to that spanned by $R$ and $L$ under a global picture; the two cases fall into distinct topological sectors.

The structure of the eigenspace deserves an intuitive explanation. The spin precession on each sublattice can be decomposed into $|\circlearrowright\rangle$ and $|\circlearrowleft\rangle$, while the sublattice degree of freedom is represented by $|A\rangle$ and $|B\rangle$. Here $|\circlearrowright\rangle$ ($|\circlearrowleft\rangle$) can be interpreted as $\text{Re}\left[(\hat{x} \pm i\hat{y})e^{i(\omega t - kx)}\right]$. With all constraints relieved, the linear space spanned by the direct products of these vectors is 4D. Under the impact of TBC, however, the magnon eigenspace only covers a 2D subspace. Specifically, the quanta of $\alpha$ and $\beta$ eigenmodes described by Eqs. (17) and (18) can be recast into a suggestive from

$$|\alpha\rangle = \tilde{\alpha}_k |0\rangle = u_{\text{large}}^\alpha |A\rangle \otimes |\circlearrowright\rangle + u_{\text{small}}^\alpha |A\rangle \otimes |\circlearrowleft\rangle - u_{\text{small}}^\alpha |B\rangle \otimes |\circlearrowright\rangle - u_{\text{large}}^\alpha |B\rangle \otimes |\circlearrowleft\rangle, \tag{23a}$$

$$|\beta\rangle = \tilde{\beta}_k |0\rangle = u_{\text{large}}^\beta |A\rangle \otimes |\circlearrowright\rangle + u_{\text{small}}^\beta |A\rangle \otimes |\circlearrowleft\rangle + u_{\text{small}}^\beta |B\rangle \otimes |\circlearrowright\rangle + u_{\text{large}}^\beta |B\rangle \otimes |\circlearrowleft\rangle, \tag{23b}$$

where $\left|u_{\text{small}}^{\alpha(\beta)}\right| < \left|u_{\text{large}}^{\alpha(\beta)}\right|$ with $u_{\text{large}}^{\alpha(\beta)} + u_{\text{small}}^{\alpha(\beta)} \propto \sqrt{q_{\alpha(\beta)}^{1(2)}}$ and $u_{\text{large}}^{\alpha(\beta)} - u_{\text{small}}^{\alpha(\beta)} \propto \sqrt{q_{\alpha(\beta)}^{2(1)}}$. Comparatively, the eigenspace under the constraint of ordinary PBC, as illustrated in Fig. 3(d), is spanned by

$$|R\rangle = \eta_{\text{small}} |A\rangle \otimes |\circlearrowright\rangle - \eta_{\text{large}} |B\rangle \otimes |\circlearrowright\rangle, \tag{24a}$$

$$|L\rangle = \eta_{\text{large}} |A\rangle \otimes |\circlearrowleft\rangle - \eta_{\text{small}} |B\rangle \otimes |\circlearrowleft\rangle, \tag{24b}$$

where $0 < \eta_{\text{small}} < \eta_{\text{large}}$ and the minus sign indicates a $\pi$ phase difference for $S_A$ and $S_B$ shown in Fig. 3(d). This is apparently different from that under the TBC. These two subspaces, in spite of their partial overlap for some special parameters, are generally distinct and cannot be transformed into each other by a linear combination of the coefficients, given that $q_\alpha^{1(2)} \neq q_\beta^{1(2)}$. Starting from a 4D Hilbert space, one needs to specify the PBC ($A \to A$ and $B \to B$) or TBC ($A \to B$ and $B \to A$) when reconciling site $i$ with $i+N$, thereby projecting the original 4D space into a chosen 2D subspace. In Sec.4, one can further appreciate the distinctions between the two different 2D subspaces.

One may wonder what happens when the system size approaches infinity. In this limit, the spectral shift of the $\alpha$ branch vanishes so the two branches become degenerate in terms of eigenvalues. Nevertheless, the eigenvectors (hence the eigenspace) for each allowed $k$ remain structurally distinct from their counterparts associated with the PBC. In other words, the eigenspace conforming with TBC does not coalesce with the eigenspace under PBC as the system approaches infinity. In particular, the cumulative $\pi$ phase shift of the $\alpha$ mode is topologically protected and independent of the system size.

Now let us inspect the manifestation of the proposed TISB. It is established that in easy-axis AFM materials such as $MnF_2$ [19,32], $R$ and $L$ are degenerate in energy in the absence of magnetic fields. In easy-plane AFM materials such as NiO, the existence of hard-axis anisotropy explicitly breaks the rotational symmetry in the spin Hamiltonian and lifts the degeneracy, rendering $L - R$ and $L + R$ the magnon eigenmodes [20–22]. In these common scenarios, the magnon eigenstates exhibit the same symmetry with the spin Hamiltonian, as we only consider the excited states while excluding the ground state that is subject to the spontaneous symmetry breaking and recasts the broken symmetry in the Goldstone modes. By contrast, the Hamiltonian (5) we adopted preserves the local rotational symmetry (with respect to the local easy axis), which is explicitly broken in the magnon eigenstates ascribing to the Möbius TBC. This intriguing phenomenon is a direct manifestation of the TISB, *i.e.*, non-trivial topology in the real space alone leads to lifted degeneracy in the eigenvalues, as well as reduced symmetry in the eigenspace of elementary excitations, without the aid of symmetry-breaking interactions. Because a quantum of angular momentum associated with the $R$-($L$-)circular mode is $+\hbar$ ($-\hbar$), the TISB we found should be followed by the suppression of longitudinal magnon spin currents on a Möbius strip.

To close this section, we mention that the standing waves formed by the $\alpha$ and $\beta$ modes can be locally excited and distinguished through a microwave. While it is difficult to fabricate and measure an AFM Möbius strip using natural materials, our findings are amenable to engineered meta-materials such as magnonic crystals made of artificial AFM units, which can be as large as centimeters with just hundreds of artificial unit cells. If we align the rf field of a microwave source with the longitudinal and transverse directions of the strip, the $\alpha$ and $\beta$ modes can then be selectively driven with different absorption rates. This is because the rf field directly couples the total spin $S$ rather than the Néel vector, and $S$ is polarized differently in these two modes [as shown in Fig. 3]. By exciting the lowest few standing-wave modes using a frequency-tunable source, the pattern of nodes shown in Fig. 4(a) should in principle be detectable through time-resolved microscopy [33] and other optical approaches.

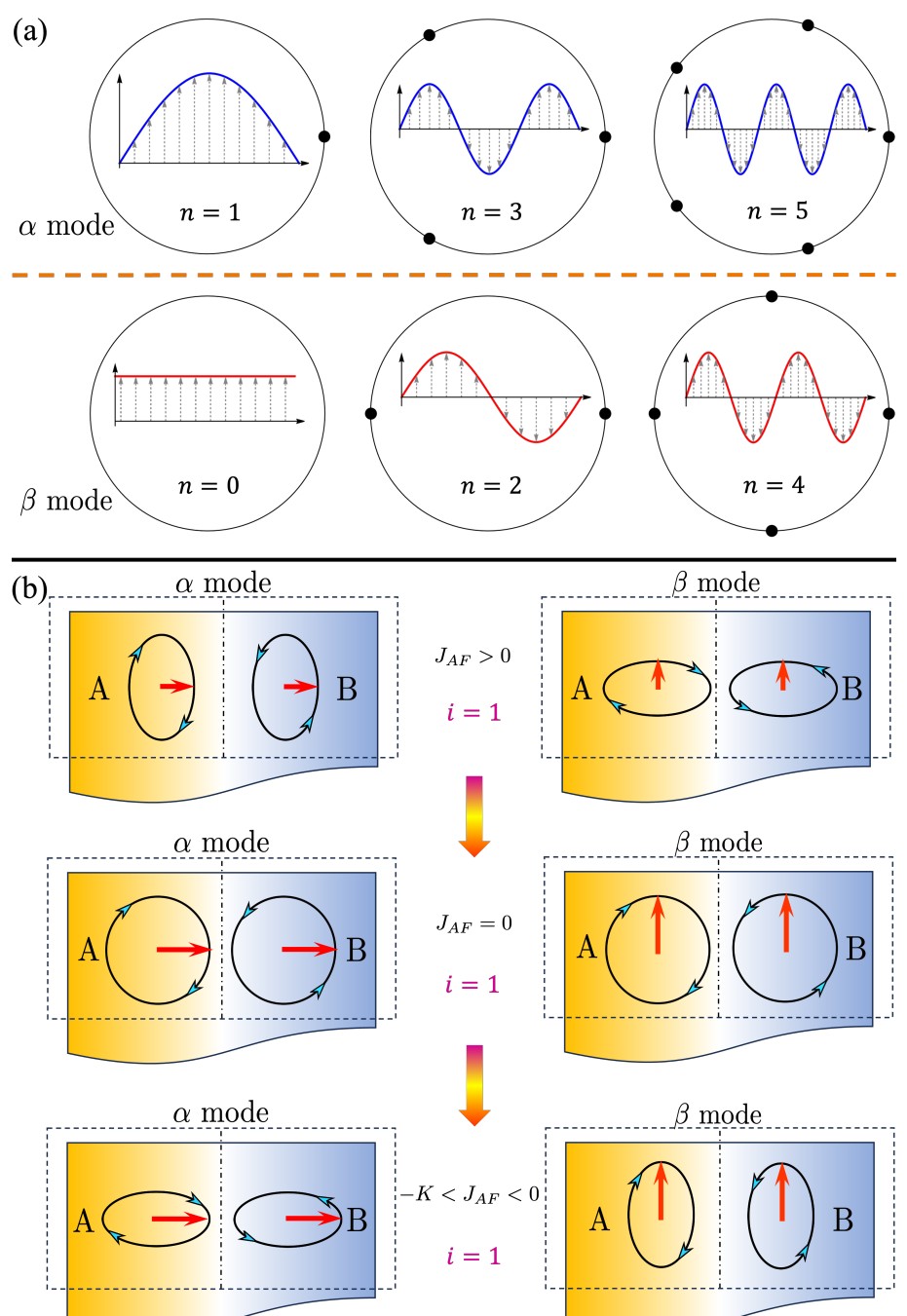

Figure 4: (a) Illustration of the standing-wave configurations for the three lowest $\alpha$ modes, each with an odd number of nodes $n$, and for the three lowest $\beta$ modes, each with an even number of nodes $n$. The vertical axis indicates the amplitude of spin oscillation, where a sign difference corresponds to a $\pi$ phase difference. (b) Illustration of the elliptical spin trajectories (on site $i = 1$) that smoothly deform when $J_{AF}$ varies. Other sites (not shown) follow a similar pattern of change.

## 4  Exotic Phase

While the major axis of the elliptical orbits for $\alpha$ modes lie in the $x$-direction (longitudinal direction of the strip), they are in the $y$-direction for the $\beta$ modes. This important property, according to Eq. (18), is attributed to $q^1_{\alpha(\beta)} > q^2_{\alpha(\beta)}$, where $q^1_{\alpha(\beta)} = Q_{\alpha(\beta)} + J_{AF}/2$ and $q^2_{\alpha(\beta)} = Q_{\alpha(\beta)} - J_{AF}/2$ with $J_{AF} > 0$. On this point, it is instrumental to notice that if $J_{AF} < 0$, *i.e.*, the AFM inter-chain exchange coupling turns ferromagnetic, then the opposite relation will be true: $q^1_{\alpha(\beta)} < q^2_{\alpha(\beta)}$, leading to elongated trajectories in the $y$- and $x$-direction for the $\alpha$ and $\beta$ modes, respectively. However, an immediate problem with the $J_{AF} < 0$ regime is that the AFM ground state may become unstable, which will be reflected as the magnon spectra (eigenvalues) touching zero.

One may naively expect a phase transition when $J_{AF}$ flips sign such that the AFM state yields to the ferromagnetic state and necessarily forms a domain wall [23, 24]. However, a careful inspection reveals something counter-intuitive. According to Eqs. (15) and (16),

$$q^1_\alpha = K + J_{AF} + J_F[1 - \cos(k - \xi\pi/L)l], \tag{25a}$$

$$q^2_\alpha = K + J_F[1 - \cos(k - \xi\pi/L)l], \tag{25b}$$

but now $J_{AF} < 0$ is assumed. At momentum $k = \xi\pi/L$, both $q^1_\alpha$ and $q^2_\alpha$ reach minimum, so does the eigenvalue:

$$\min[\omega_\alpha] = S\sqrt{q^1_\alpha\left(\frac{\xi\pi}{L}\right)q^2_\alpha\left(\frac{\xi\pi}{L}\right)} = S\sqrt{K(K + J_{AF})}, \tag{26}$$

and similarly, the minimum of $\omega_\beta$ occurs at $k = 0$, where $\min[\omega_\beta] = \min[\omega_\alpha]$. One can tell from Eq. (26) that the AFM ground is preserved even for $-K < J_{AF} < 0$. A phase transition must take place at $J_{AF} = -K$, where magnons become infinitely soft and proliferate, melting the AFM configuration and driving the ground state into a ferromagnetic domain wall (note that a uniform ferromagnetic configuration is forbidden by the Möbius boundary condition).

The surprising region of $-K < J_{AF} < 0$ features an exotic phase in which the AFM ground state can stand even though the inter-chain and intra-chain exchange interactions are both ferromagnetic. As schematically illustrated in Fig. 4(b), the elliptical trajectories of sublattice spins deform smoothly while $J_{AF}$ varies from positive to negative, until the threshold $J_{AF} = -K$. An ideal circular polarization for each sublattice is reached at $J_{AF} = 0$. In the exotic regime $-K < J_{AF} < 0$, the principal axes of spin precession for both the $\alpha$ and $\beta$ modes indeed swap relative to the $J_{AF} > 0$ case. Nonetheless, the magnon eigenspace is still a disparate subspace comparing to that dictated by the PBC, regardless of the sign of $J_{AF}$.

The $\alpha$ and $\beta$ modes in this exotic phase has a unique feature which is fundamentally impossible for the conventional linearly-polarized modes $L + R$ and $L - R$: The trajectories of $S_A$ and $S_B$, when being projected onto the local $x - y$ plane, meet along the major axis and become opposite to each other along the minor axis. While the linear polarization status of the Néel vector and the total spin vector remain the same for both $J_{AF} > 0$ and $-K < J_{AF} < 0$, the magnitudes of their oscillations are noticeably different in the two phases, which can be pictorially inferred from Fig. 4(b) and be rigorously quantified via the same set of formulas in Sec. 2 and 3.

## 5  Conclusion and Outlook

In summary, we have proposed the TISB mechanism arising from the difference between PBCs and TBCs. As a concrete example, we apply TISB to antiferromagnetic magnons on a Möbius

strip: the TBCs dictate that the Néel vector is linearly polarized in all eigenmodes, lifting the usual degeneracy of the circular-polarization modes, hence obviating the chirality of the order parameter dynamics. The true eigenmodes not only depart from conventional magnon branches but also give rise to unconventional standing-wave patterns. Moreover, the Möbius topology stabilizes an exotic antiferromagnetic phase even when the interchain exchange coupling turns ferromagnetic.

Even though we have demonstrated the TISB in the AFM magnons on a Möbius strip, the mechanism itself is general and can manifest in other contexts with different quasiparticles or TBCs. For example, we anticipate that the eigenmodes of phonon excitations on a Möbius strip to be linearly polarized while the circularly-polarized chiral phonons are suppressed by the Möbius topology. For photons that are governed by Maxwell's equations, the axis of circular polarization is parallel to the momentum $\boldsymbol{k}$, which defines a distinct geometry compared to ours, calling for a separate investigation. For tight-binding electrons on a Möbius strip, the wavefunction is a two-component spinor $\Phi(\boldsymbol{r}) = [\phi_\uparrow(\boldsymbol{r}), \phi_\downarrow(\boldsymbol{r})]^{\mathrm{T}}$, and the TBC in Sec. 1.2 becomes: $\hat{T}_L \Phi(x, y) = -\Phi(x + L, -y)$. While the Hamiltonian can formally parallel our magnon model, the fermionic statistics also brings a fundamental distinction in the diagonalization [which does not involve $\sigma_z$ as that in obtaining Eq. (15)]. Consequently, the precise impact of TBCs on electronic bands also merits a dedicated, detailed study.

Under alternative TBCs such as a Klein bottle, even the AFM magnons could acquire new features of TISB beyond what we have shown in this work. Therefore, our findings could greatly inspire a broader research endeavor in the near future highlighting the profound impact of real-space topology on the physical nature of elementary excitations.

## Acknowledgements

We thank Qian Niu for helpful discussions.

**Funding information**    This work is supported by the Air Force Office of Scientific Research under Grant No. FA9550-19-1-0307 and the UC Regents' Faculty Development Award.

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
