# Peer review of "Topology-induced symmetry breaking demonstrated in antiferromagnetic magnons on a Möbius strip"

_SciPost Physics, doi:SciPost Phys. 19, 048 (2025)_

## Round 1 · Referee Report · Anonymous (Referee 1) · 2025-6-29

Strengths
-
The paper reports on an interesting observation, viz. the topology-induced breaking of symmetry in the excitation spectrum of a magnetic model on a Möbius strip that occurs despite the lack of such symmetry-breaking in the ground state.
-
The paper is very clearly written, with ample and helpful interpretation of the results obtained.
Weaknesses
-
The paper has an 'outlook' section, but no 'conclusion' section where the results are summarised. Such a conclusion section is required by SciPost Physics general acceptance criterion 4.
-
There are occasional errors in the grammar, e.g. "topological protected" on page 7.
-
The abbreviative convention in which a second sentence is included in parentheses within a first, rather than written separately, is overused, sometimes to the point of impairing meaning. See, for example, "$q^1_{\alpha(\beta)} < q^2_{\alpha(\beta)},$ leading to elongated trajectories in the $y$-direction ($x$-direction) for the $\alpha$ ($\beta$) modes" on page 7. This technique makes things easier for the writer, but harder for the reader; I recommend avoiding its use altogether.
Report
The general acceptance criteria are also met, with one exception, number 4: the paper does not currently have "a clear conclusion summarizing the results". However, I don't think it would be difficult for the authors to add one.
I therefore recommend publication, subject to the requirement that a conclusion be written, and the recommendation that weaknesses 2 and 3 in the above list are also remedied.
Requested changes
- Add a conclusion.
- Correct grammatical errors.
- Remove use of the "We see A (B) in case C (D)" device.
Recommendation
Publish (meets expectations and criteria for this Journal)
Reply to Referee 1
We thank you for evaluating our manuscript, which we believe now meets the SciPost Physics criteria. Below, we address your concerns.
Referee 1: The paper has an "outlook" section, but no "conclusion" section where the results are summarised. Such a conclusion section is required by SciPost Physics general acceptance criterion 4.
Our response: Now we have added the conclusion into the last section "Conclusion and Outlook".
Referee 1: There are occasional errors in the grammar, e.g. "topological protected" on page 7.
Our response: We have corrected the errors and typos in the manuscript shown with colored words.
Referee 1: The abbreviative convention in which a second sentence is included in parentheses within a first, rather than written separately, is overused, sometimes to the point of impairing meaning. See, for example, "$q_{\alpha(\beta)}^1$ < $q_{\alpha(\beta)}^2$, leading to elongated trajectories in the y-direction (x-direction) for the
$\alpha\,(\beta)$ modes" on page 7. This technique makes things easier for the writer, but harder for the reader; I recommend avoiding its use altogether.
Our response: The original idea was to use this abbreviative convention to prevent the redundancy. Following your suggestion, now we have significantly reduced the use of this form as much as we can for easier reading. Some of the sentences are rewritten.

Author: Kuangyin Deng on 2025-07-23 [id 5667]
(in reply to Report 2 by Daniel Cabra on 2025-07-13)Please see the response in the attached file.
Attachment:
SciPost_reply_to_referee_2.pdf

---

## Round 1 · Referee Report · Daniel Cabra (Referee 2) · 2025-7-13

**Referee Report on "Topology-induced symmetry breaking demonstrated in antiferromagnetic magnons on a Möbius strip" by Deng and Cheng**

In this work, the authors explore the influence of real-space topology, specifically, that of the Möbius strip, on the eigenmodes (magnons). By considering a model composed of two ferromagnetic spin chains coupled antiferromagnetically and arranged into a Möbius strip, they demonstrate that topological boundary conditions (TBCs) can induce a form of symmetry breaking in the excitation spectrum, which they name **topology-induced symmetry breaking (TISB)**.

The key result is that despite the Hamiltonian preserving local O(2) spin rotational symmetry, the Möbius topology forces all magnon eigenmodes to be **linearly polarized**, eliminating chirality and leading to a split into two non-degenerate branches with distinctive standing wave patterns (one supporting half-integer modes, the other integer modes). This effect persists even in the absence of curvature and is fundamentally distinct from spontaneous symmetry breaking of the ground state.

The concept of *topology-induced symmetry breaking* is, to the best of my knowledge, novel and presents a new way of thinking about how boundary conditions can modify the symmetry of excitations.

Importantly, the work goes beyond superficial analogies and develops a fully microscopic model (Heisenberg Hamiltonian), applying analytical methods to obtain and interpret the magnon modes. The idea that Möbius topology alone can eliminate chirality from spin wave excitations, even when chirality would otherwise be protected by local symmetry, is both original and of general interest. The results are potentially applicable to engineered meta-materials (e.g. magnonic crystals).

The term "topology-induced symmetry breaking" (TISB) is compelling but perhaps deserves a clearer definition upfront, with a contrast to spontaneous symmetry breaking in the introduction.

The authors correctly handle the antiferromagnetic spin wave theory using Holstein–Primakoff and Bogoliubov transformations, carefully adapting them to Möbius TBCs. They identify that the Möbius boundary condition mixes sublattices and prevents the eigenmodes from being globally circularly polarized.

The "exotic phase" with **ferromagnetic interchain couplings** but preserved AFM order is a particularly intriguing prediction. The analysis of the soft mode and stability threshold is convincing. Could there be **experimental observables** that directly detect the spectral shift or suppressed chirality, beyond the proposed time-resolved microscopy? A brief discussion of **fermionic systems**, such as electrons in a Möbius ring, is mentioned in the outlook but not developed. A qualitative contrast with the magnon case would enrich the discussion.

The manuscript is clearly written, physically insightful, and the theoretical results are carefully derived and clearly illustrated.

I **recommend acceptance with minor revisions**. The manuscript meets all criteria for *SciPost Physics*, including scientific novelty, clarity, and relevance. The results are solid, the idea is original, and the implications merit publication.

Minor comment: In several places, "eigenmodes of excitation" could simply be "eigenmodes" or "excitations" for conciseness.

---

## Round 2 · List of Changes

1. We have modified the last section as “Conclusion and Outlook” and added descriptions for fermions.
  2. We have corrected some errors and typos in the manuscript.
  3. we have significantly reduced the use of double brackets abbreviative convention in the main text as much as we can for easier reading. Some of the sentences are rewritten.
  4. At the end of Sec. 1.2, we have modified our phrases with a clear definition of TISB and a contrast to pontaneous symmetry breaking.

---

## Editorial Decision

published